# Wing bone geometry reveals active flight in *Archaeopteryx*

Dennis F.A.E. Voeten[1,2], Jorge Cubo[3], Emmanuel de Margerie[4], Martin Röper[5,6], Vincent Beyrand [1,2], Stanislav Bureš[2], Paul Tafforeau [1] & Sophie Sanchez [1,7]

*Archaeopteryx* is an iconic fossil taxon with feathered wings from the Late Jurassic of Germany that occupies a crucial position for understanding the early evolution of avian flight. After over 150 years of study, its mosaic anatomy unifying characters of both non-flying dinosaurs and flying birds has remained challenging to interpret in a locomotory context. Here, we compare new data from three *Archaeopteryx* specimens obtained through phase-contrast synchrotron microtomography to a representative sample of archosaurs employing a diverse array of locomotory strategies. Our analyses reveal that the architecture of *Archaeopteryx*'s wing bones consistently exhibits a combination of cross-sectional geometric properties uniquely shared with volant birds, particularly those occasionally utilising short-distance flapping. We therefore interpret that *Archaeopteryx* actively employed wing flapping to take to the air through a more anterodorsally posteroventrally oriented flight stroke than used by modern birds. This unexpected outcome implies that avian powered flight must have originated before the latest Jurassic.

[1] European Synchrotron Radiation Facility, 71 Avenue des Martyrs, CS-40220, 38043 Grenoble Cedex, France. [2] Department of Zoology and Laboratory of Ornithology, Palacký University, 17. listopadu 50, 771 46 Olomouc, Czech Republic. [3] Sorbonne Université, CNRS-INSU, Institut des Sciences de la Terre Paris, ISTeP UMR 7193, F-75005 Paris, France. [4] CNRS, Laboratoire d'éthologie animale et humaine, Université de Rennes 1, Université de Caen Normandie, 263 Avenue du Général Leclerc, 35042 Rennes, France. [5] Bürgermeister-Müller-Museum, Bahnhofstrasse 8, 91807 Solnhofen, Germany. [6] Bayerische Staatssammlung für Paläontologie und Geologie, Richard-Wagner-Str. 10, D-80333 München, Germany. [7] Science for Life Laboratory and Uppsala University, Subdepartment of Evolution and Development, Department of Organismal Biology, Evolutionary Biology Centre, Norbyvägen 18A, 752 36 Uppsala, Sweden. Correspondence and requests for materials should be addressed to D.F.A.E.V. (email: dennis.voeten01@upol.cz)

The earliest phases of avian evolution and development of avian flight remain obscured by the rarity of representative fossil material and consequential limited phylogenetic resolution[1]. As the oldest potentially free-flying avialian known[1–3], *Archaeopteryx* represents the prime candidate to consider in resolving the initial chapter of bird flight. Although the traditional dichotomy between an arboreal and a cursorial origin of avian flight[4] has relaxed towards the consideration of intermediate perspectives[3,5], the question whether the first flying bird-line dinosaurs took flight under their own power remains unanswered.

Skeletal adaptations that structurally accompany known locomotor modes provide reliable proxies for inferring the habits of extinct tetrapods. The cross-sectional geometry of limb bones is largely determined by evolutionary selection on the interplay between strength and weight[6] and continuous morphological and structural adaptation to the biomechanical loading regimes experienced during life[7]. Therefore, the avian wing skeleton informs on this stress regime through the application of beam theory mechanics[8–10].

Although the value of exceptional and rare fossils discourages physical cross-sectioning, Propagation Phase-Contrast Synchrotron X-Ray Microtomography (PPC-SRμCT) now offers non-destructive alternatives[11]. Using PPC-SRμCT at the European Synchrotron Radiation Facility with a novel data acquisition protocol (Supplementary Note 1), we visualised complete circa mid-diaphyseal humeral and ulnar cross sections of three *Archaeopteryx* specimens (Fig. 1a–h) because these elements exhibit the strongest flight-related biomechanical adaption in the modern avian brachium[10,12]. Their full transverse cross-sectional geometry was reconstructed (Fig. 1i–n) and compared with an extensive set of archosaurian humeri and ulnae representing 69 species spanning a wide variety of locomotory behaviours (Supplementary Fig. 1 and Supplementary Data 1). Notably, we

included the basal "long-tailed" pterosaur *Rhamphorhynchus* and the derived "short-tailed" pterosaur *Brasileodactylus* in our archosaurian reference set to contrast conditions associated with pterosaurian volancy against those of the independently arisen avian flight apparatus. Although the pterosaurian and avian flight apparatus differ in fundamental morphological aspects, comparing them may be expected to reveal underlying analogous adaptations in wing bone geometry.

Raw virtual slice data revealed that the long-bone cortex of *Archaeopteryx* exhibits a vascular density in the range of modern birds, which proposes substantial metabolic performance. Cortical vascular density strongly varies between two specimens of *Archaeopteryx* studied, which we interpret, based on body size differences, to reflect ontogenetic disparity. Relative cortical thickness of archosaurian anterior limb bones successfully discriminates between known non-volant and volant forms, and confidently indicates that *Archaeopteryx* was volant. Mass-normalised torsional resistance in the same set of limb bones describes a gradient within modern volant birds that ranges from flight strategies relying on occasional or intermittent flapping to gain altitude to hyperaerial specialists employing prolonged gliding or soaring in their flight. The three specimens of *Archaeopteryx* were found to unanimously ally with birds that incidentally employ flapping flight to evade predators or cross physical barriers, through which we interpret that *Archaeopteryx* actively used its wings to take to the air. Since the morphology of the flight apparatus in *Archaeopteryx* is known to be incompatible with the flight stroke executed by modern volant birds, we furthermore conclude that *Archaeopteryx* adopted a different flight stroke than used by birds today. Finally, we found that the evolution from primitive long-tailed pterosaurs to more derived short-tailed pterosaurs was accompanied by qualitatively comparable modifications to wing bone geometry as those that distinguish

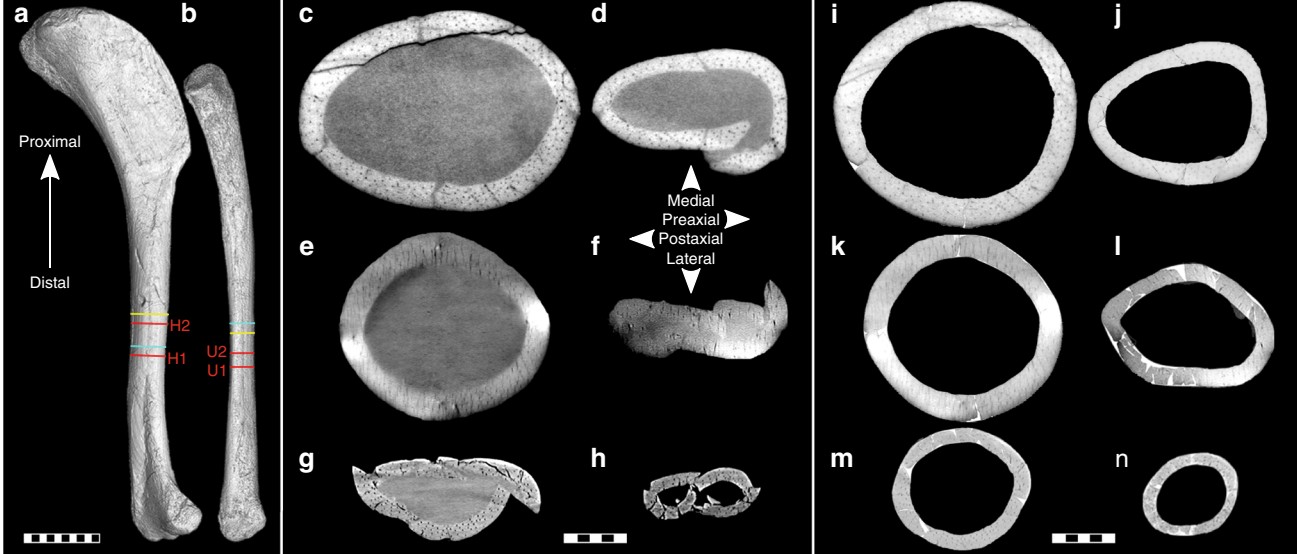

**Fig. 1** *Archaeopteryx* humeral and ulnar virtual cross sections used in this study. **a** Right humerus and **b** right ulna of the ninth (Bürgermeister–Müller) specimen in lateral, respectively, medial view, with virtual sampling locations (red) and relative sampling locations in seventh (Munich) specimen (light blue) and fifth (Eichstätt) specimen (yellow). **c–h** Virtual cross sections, as preserved, with optimised intraosseous contrast, of (**c**) right humerus (H2) and (**d**) right ulna (U2) of ninth specimen, (**e**) right humerus and (**f**) right ulna of seventh specimen, and (**g**) left humerus and (**h**) left ulna of fifth specimen. **i–n** Reconstructed cross-sectional geometry, with optimised contrast of bone margins, of (**i**) humerus and (**j**) ulna of ninth specimen, (**k**) humerus and (**l**) ulna of seventh specimen, and (**m**) humerus and (**n**) ulna of fifth specimen; pure white indicates supplemented fragments. Morphological orientation applies to all sections (**c–h**). Scale bar (**a**, **b**) 10 mm; scale bars (**c–n**) 1 mm

*Archaeopteryx* and principally flapping birds from hyperaerial birds, respectively.

## Results

**Cortical vascularisation**. Contrary to previously reported data[13], the bone cortex of *Archaeopteryx* is well vascularised (Fig. 1c–h, Supplementary Data 1). The ninth specimen exhibits an average cortical vascular density (Supplementary Fig. 2) in the lower range of neognaths (circa 69 canals/mm$^2$), but higher than the hoatzin (*Opisthocomus hoazin*; circa 43 canals/mm$^2$), whereas the average cortical vascular density of the smaller fifth specimen is higher than in most neognaths assessed (circa 116 canals/mm$^2$; Supplementary Fig. 3). This is consistent with a higher bone growth rate[14] and associated higher resting metabolic rate[15] than inferred from nearly avascular bone chips of the seventh specimen[13]. Ontogenetic progression is accompanied by a reduction in the apposition rate and vascular density of forming bone[16]. Providing that the fifth and ninth specimen represent the same species or that we are observing the shared generic ontogenetic pattern, their marked differential vascular density scaling inversely with body size would indicate disparate ontogenetic stages for these individuals.

**Relative cortical thickness**. Average relative cortical thickness of anterior limb bones successfully separates volant from non-volant archosaurs in our data set, although individual element values may slightly cross the average relative cortical thickness value of 0.60 found to separate these groups (Supplementary Figs. 4 and 5). Within non-volant groups, aquatic and (facultatively) quadrupedal species have relatively thicker humeral bone cortices than terrestrial bipeds. The basal pterosaur *Rhamphorhynchus* exhibits an average relative cortical thickness in the upper range of volant archosaurs, whereas the pterodactyloid pterosaur *Brasileodactylus* presents the lowest relative cortical thickness recorded. Only volant birds that engage in wing-propelled diving may exhibit an average relative cortical thickness in the range of non-volant archosaurs to counteract buoyancy and manage the demands of subaqueous locomotion[17]. Charadriiformes (Supplementary Fig. 1) share an elevated average relative cortical thickness with respect to other orders of flying birds (Supplementary Data 1, Supplementary Fig. 6), likely as an adaptation to negotiate "strong winds" in coastal and marine habitats[10]. The humeral and ulnar bone walls of *Archaeopteryx*, comparatively thinner than those of any element in the non-volant sample, reveal a strong affinity with volant birds but disqualify this taxon as a wing-propelled diver[18] or obligatory "wind-assisted" flyer[19].

**Mass-normalised torsional resistance**. Because a lower relative cortical thickness positions bone material further away from the bone section centroid (Supplementary Fig. 2) than a higher relative cortical thickness at the same amount of bone present, relative cortical thickness and mass-normalised torsional resistance are inherently not completely decoupled. Furthermore, mass-normalised torsional resistance retains a small yet significant allometric effect that reflects the inherent proportional relation between flight adaptations and body size, and should thus not be removed when investigating the locomotory affinities of extinct taxa[12,20]. Nevertheless, we focus on obvious signals and trends that exist relative to such relations, since those are particularly informative towards distinguishing the effects of related locomotory regimes.

Mass-normalised torsional resistance successfully separates non-avian theropods from flightless birds with comparable body mass values, but also exhibits subtle variation across avian flight modes (Supplementary Figs. 4 and 5). Burst-flying[20] birds (incidental explosive take off and brief horizontal flight followed by a running escape) exhibit humeral and ulnar relative torsional resistance values overlapping with those of intermittent bounding[20] flyers (flapping phases aimed at gaining altitude and speed, alternated with passive phases with folded wings). However, burst-flying[20] birds attain body mass values that are, on average, two orders of magnitude higher than those of intermittent bounding[20] flyers, which is informative when discriminating these two groups. Conversely, flap-gliding[20] birds have a similar to higher humeral and ulnar relative torsional resistance compared to burst[20]-adapted and most continuously flapping[20] flyers at body masses that are, on average, one order of magnitude lower. Notably, the two large non-domesticated anatids in our data set share elevated relative torsional resistance values compared to other continuously flapping birds. Soaring birds[20] may attain comparatively high body mass values, yet exhibit distinctly elevated normalised torsional resistance values relative to their body mass throughout (Supplementary Figs. 4 and 5).

The fifth, seventh, and ninth specimens of *Archaeopteryx* have reconstructed body mass values of 158, 254 and 456 g[13], respectively. These inferred ontogenetic[13] mass differences do not scale linearly with humeral and ulnar torsional resistance: the seventh and ninth specimen exhibit comparable mass-corrected torsional resistance values that are elevated proportional to those of the fifth specimen and within the lower range of modern volant birds. This may reflect an ontogenetic ecomorphological shift between the ages associated with reconstructed body mass values of 158 and 254 g towards increased volant adaptation. The seventh and ninth specimens of *Archaeopteryx* exhibit relative humeral torsional resistance approaching those of modern short[21]/burst[20] flying birds of similar mass and higher than some heavier non-volant archosaurs. Ulnar torsional resistance values in these specimens are comparable to those of lighter volant birds and much heavier non-volant birds, and are higher than in the small non-avian coelurosaur *Compsognathus*. A shared proportional disparity between relative humeral and ulnar torsional resistance in the seventh and ninth specimen of *Archaeopteryx* with respect to flying birds indicates an underlying different employment of the epipodium, such as a possible larger contribution of the radius in wing load transfer to the humerus relative to the modern avian flight apparatus. Like *Archaeopteryx*, humeral and ulnar relative torsional resistance of *Rhamphorhynchus*, circa 40% lighter than the fifth specimen of *Archaeopteryx*, also scale with the lightest volant birds that have body masses up to one order of magnitude lower. The piscivorous diet of *Rhamphorhynchus*[22] strongly favours active flight, and specimens substantially smaller than the individual considered here have been concluded to have been volant[23]. This, in turn, demonstrates that comparatively low relative torsional resistance of the bones supporting the limb carrying the airfoil does not preclude the capacity of active flight.

**Volancy**. Phylogenetic Principal Component Analysis (pPCA) of the referred parameters places *Archaeopteryx* in a domain shared almost exclusively with modern volant birds (Supplementary Fig. 7). Partitioning Around Medoids (PAM) of the first three phylogenetic Principal Components resolves volancy with a 95.52% success rate and allies *Archaeopteryx* with volant archosaurs (Supplementary Data 2). Only three volant but (incidentally) wing-propelled diving birds were misclassified as flightless through their elevated relative cortical thickness. This outcome, including misclassification of the three wing-propelled divers and

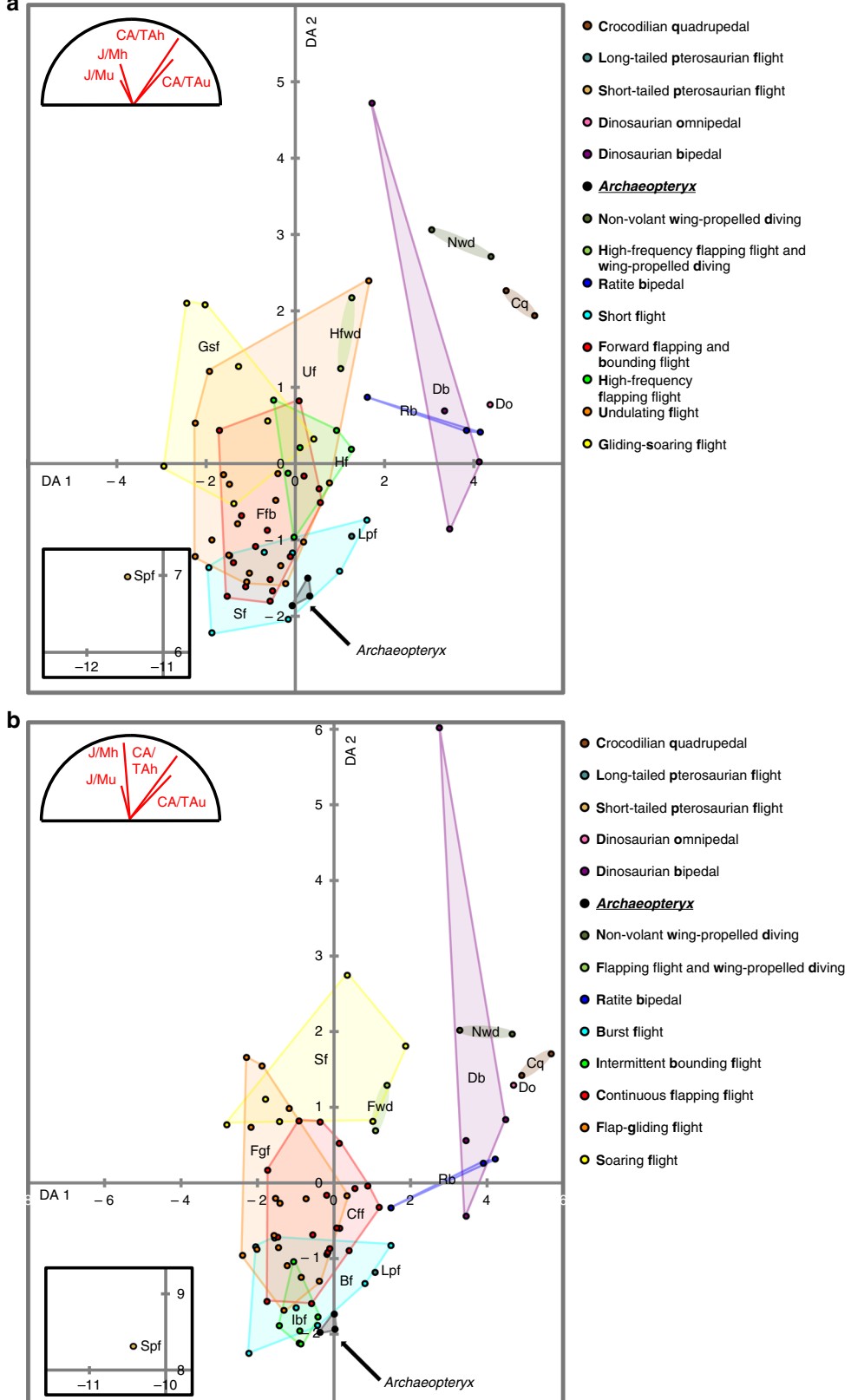

**Fig. 2** LDA plot for specific archosaurian humeral and ulnar CA/TA and J/M. First and second linear discriminant axes are presented. Classification follows the locomotory divisions adapted from **a** Viscor et al.[21] and **b** Close et al.[20], non-pterosaurian flight strategies represent avian flight modes. Dots correspond to species, *Archaeopteryx* specimens plotted individually. Coloured hulls delimit groups with a minimum of three representatives; coloured ellipses link the members of groups with two representatives. Parameters labelled ¨_h¨ and ¨_u¨ in loading biplots designate humeral and ulnar affinity, respectively

the assignment of *Archaeopteryx* to the volant group, is identically recapitulated by k-means clustering of the raw parameter values into two clusters (Supplementary Data 2). Additional discussion of volancy in *Archaeopteryx* through individual parameters is included in Supplementary Note 2.

**Locomotor mode**. Phylogenetic autocorrelation was found to be insignificant towards Phylogenetically Informed Discriminant Analysis (Supplementary Fig. 8), thus rendering Linear Discriminant Analysis (LDA) appropriate for resolving the locomotory affinity of *Archaeopteryx*. Linear discriminant analysis (LDA) principally separates volant and non-volant archosaurs along discriminant axis (DA) 1 (Fig. 2 and Supplementary Fig. 9) through predominantly humeral and ulnar relative cortical thickness. DA 2 is loaded more equally on all parameters than DA 1, but slightly stronger on relative cortical thickness for the classification expanded from Viscor et al.[21], and slightly stronger on mass-normalised torsional resistance for the classification expanded from Close et al.[20]. Within modern volant birds, an overlapping succession of avian flight modes extends upwards along DA 2 (Fig. 2). This sequence starts at the bottom of DA 2 with short[21] or burst[20] flight and intermittent bounding[20]. These flight modes share a strong climbing component during powered flight phases, where intermittent bounding remains restricted to species with an adult body mass typically below 200 g[21]. Forward flapping[21], high-frequency flapping[21], or continuous flapping[20] flight is observed in birds that maintain level flapping flight after take-off. The avian sequence terminates in undulating[21] or flap-gliding[20] fliers that structurally alternate between flapping and gliding during flight, and aerial specialists in the gliding–soaring[21] and soaring[20] flight categories that harvest atmospheric movements to gain altitude and engage in sustained gliding. Across avian flight strategies and relative to mass-specific effects, relative torsional resistance of wing bones appears to scale inversely with a reliance on flapping during flight. This may reflect either a multi-directional stress regime or elevated torsional loading experienced during gliding and soaring as opposed to the directionally confined stresses associated with flapping[12]. A coupled decrease in relative cortical thickness and torsional resistance of anterior limb bones accompanied the transition from non-avian dinosaurs to birds (Fig. 2). A subsequent increase in relative torsional resistance at rather constant relative cortical thickness within volant birds accompanies the establishment of distinct avian flight modes (Fig. 2). Although basal and derived pterosaurs are represented by only one taxon each, their reciprocal relationship in pPCA and LDA morphospace qualitatively agrees with those between (occasionally) flapping birds and avian hyperaerial soarers, respectively. From a biomechanical point of view, this lends some support for an analogous evolutionary trajectory leading from flapping volancy in *Rhamphorhynchus* to an affinity with prolonged soaring in *Brasileodactylus*.

Discriminant classification unanimously groups the specimens of *Archaeopteryx* with short[21] or burst[20] flyers in our set (Supplementary Data 2). Substantial group overlap within the volant avian cluster in discriminant morphospace results in relatively low percentages of correctly classified training species (53.62% and 56.52% for the Viscor et al.[21] and Close et al.[20] divisions, respectively), and this uncertainty undoubtedly carries over to the characterisation of extinct taxa with unknown locomotory strategies. Nevertheless, the position of *Archaeopteryx* is chiefly shared with short[21]/burst[20] flyers (Fig. 2 and Supplementary Fig. 9), which supports the resolved affinity with these near-synonymous flight groups. Although *Archaeopteryx*

plots close to intermittent bounding[20] birds as well, its reconstructed adult body mass[13] vastly exceeds the mass threshold to which modern bounding birds adhere[21]. Finally, short[21] and burst[20] flyers exhibit significantly deviating means from respectively gliding–soaring[21], and flap-gliding[20] and soaring[20] birds (Supplementary Tables 1 and 2), thereby underlining an affinity to habitual flapping for *Archaeopteryx*.

## Discussion

Obligatory gliding (non-serpent) amniotes, such as the extant flying lizard *Draco* and flying squirrels in the family Pteromyini, converge on the employment of typically low-aspect-ratio limb-supported patagia along the lateral body wall that are occasionally supplemented with webbed digits[24]. The pterosaurian and chiropteran flight apparatus superficially resemble derived, powered modifications of this general configuration that achieved active flight. In contrast, even the oldest avian wings represent specialised anterior limbs that are inherently mobile and structurally connect to the body exclusively through articulation with the glenoid. Such a particular modification of the non-avian maniraptoran arm, which constitutes a highly dexterous limb in its own right[25], sharply disagrees with the conventional condition shared by limbed amniotes primarily adapted to passive gliding. This advocates the employment of an actively moved wing in *Archaeopteryx*.

Earlier conclusions that *Archaeopteryx* was capable of active flight[26] have not received universal support, largely because three skeletomorphological conditions considered essential for a functional avian flight stroke (ossified, keeled sternum; supracoracoideal "pully" arrangement; glenohumeral tolerance permitting supradorsal humeral abduction) were not yet present in *Archaeopteryx*[26–30]. Such challenges in reconciling *Archaeopteryx*'s dromaeosaurid-like pectoral morphology[30] with the modern avian dorsoventral flight stroke exemplify that avian powered flight may have worked through alternative configurations in the past. A putative aerodynamic control function for the long, stiff, frond-feathered tail[2] and hindlimb plumage[31] argue for an alternative aerial posture compared to modern birds. *Archaeopteryx*'s large coracoids[30] and robust, flattened and more dorsally positioned furcula lacking hypocleidial communication with the sternum[2] could have provided support for an anterodorsally-posteroventrally oriented flight stroke cycle that was morphologically closer to the "grabbing" motion of maniraptorans[25] and did not or hardly extend over the dorsum.

The avian nature of *Archaeopteryx*'s humeral and ulnar cross-sectional geometry shares more flight-related biomechanical and physiological adaptations with modern volant birds than previously known, which we argue to reflect the shared capability of powered flight. *Confuciusornis* from the Early Cretaceous of China also lacked the supracoracoideal pully[30] and sufficient dorsal humeral excursion[5] to permit a modern avian flight stroke. However, a variety of Early Cretaceous enantiornitine and euornithine birds (Supplementary Fig. 1) was likely already capable of executing a dorsoventral wingbeat cycle[3,5,30], which suggests that the development of dorsoventral flapping is primitive for Ornithothoraces and approximately coincided with the appearance of the avian alula[3]. The origin of the modern avian flight stroke was conceivably promoted by selective pressure towards vertical take-off[30], which contributed to the prosperous avian radiation that continued ever since.

## Methods
**Materials**. Specimens of *Archaeopteryx* in this study are designated through a commonly used numerical sequence that roughly corresponds to their succession of discovery[2]. The fifth specimen of *Archaeopteryx*[2] (JM 2257) is a nearly complete and largely articulated skeleton of the smallest *Archaeopteryx* specimen known to

date. It is also known as the Eichstätt Specimen and housed at the Jura Museum in Eichstätt, Germany (JM). The seventh specimen of *Archaeopteryx*[2] (BSP 1999 I 50) is represented by a comparably complete skeleton exhibiting a substantial degree of articulation. It is formally named Solnhofen-Aktien-Verein Specimen but generally referred to as Munich Specimen, and is kept at the Paläontologisches Museum München in Munich, Germany (PMM). Skeletal elements of both the fifth and seventh specimen of *Archaeopteryx* have experienced brittle deformation during post-depositional compaction that resulted in splintering of the bone cortex. The ninth specimen of *Archaeopteryx*[2] (BMMS-BK1a) preserves a partially disarticulated right wing skeleton of comparably large size that is presently housed at the Bürgermeister–Müller–Museum in Solnhofen, Germany (BMM). It is officially named "Exemplar der Familien Ottman & Steil", also known as Bürgermeister–Müller Specimen, and colloquially referred to as "Chicken Wing". Although a certain degree of post-depositional compaction is evidenced by the presence of several fractures that propagate through the long bone cortex, cortical splintering has not occurred. Its elements have therefore largely preserved their original three-dimensional geometry[2].

Comparative material (Supplementary Data 1) was sourced from the collections of the European Synchrotron Radiation Facility, Grenoble, France (ESRF), the Musée des Confluences, Lyon, France (MdC), the Muséum national d'Histoire naturelle, Paris, France (MNHM), the Museum of Evolution, Uppsala, Sweden (MoE), and the University of Manchester, Manchester, England (TUoM).

**Data acquisition.** The humeral and ulnar cross-sectional geometry of the three specimens of *Archaeopteryx*, 28 species of neornithine birds, the small coelurosaur *Compsognathus longipes*, the rhamphorhynchid pterosaur *Rhamphorhynchus* sp., the anhanguerid pterosaur *Brasileodactylus araripensis*, and the crocodile *Crocodylus niloticus* were visualised through propagation phase-contrast synchrotron X-ray microtomography (PPC-SRµCT) at beamlines BM05 and ID19 of the European Synchrotron Radiation Facility. An ulnar cross section of aff. *Deinonychus antirrhopus* was imaged and subsequently paired with a humeral section of *D. antirrhopus* from literature[32] through morphological and dimensional comparison.

Synchrotron X-ray tomography was conducted by utilising an optimised polychromatic beam with sufficient coherence to permit the application of PPC-SRµCT. Propagation phase-contrast imaging relies on a certain propagation distance between the sample and the detector that allows for the exploitation of the phase-contrast effect towards emphasising low-contrast features[33]. The fifth and seventh specimen of *Archaeopteryx* were imaged in accumulation mode, a novel acquisition protocol developed for imaging fossils encased in lithic slabs. The motivation for and implementation of the accumulation mode are explained in Supplementary Note 1 Further details of the adopted data acquisition parameters for each sample are provided in Supplementary Data 3.

**Data processing.** Three-dimensional volume reconstruction was conducted through filtered back projection following a phase retrieval protocol that relies on a homogeneity assumption by using a modified[33] version of the algorithm developed by Paganin et al.[34]. Virtual two-dimensional cross-sectional slides were extracted directly from the reconstructed volumes at the developmental mid-diaphyseal plane oriented perpendicular to the local bone long axis in VGStudio MAX 2.2 (Volume Graphics, Heidelberg, Germany). Avian ulnae that carry a quill knob (ulnar papilla) at mid-diaphysis were virtually sampled at the level nearest to mid-diaphysis where no quill knob was present. The data set was supplemented with avian samples used in earlier studies[8,10,35]. Furthermore, scaled figures depicting complete perpendicular humeral and ulnar cross sections of avian and non-avian archosaurs were sourced from literature[17,36–39] and processed in tandem with data obtained through PPC-SRµCT.

**Cross-sectional geometry.** Based on the characteristics unfolded nature of the Solnhofen Plattenkalk[40], the geometry of *Archaeopteryx* wing elements was assumed to have experienced only brittle deformation during unidirectional compaction with insignificant movement of bone fragments perpendicular to the visualised cross sections. Two-dimensional restoration was conducted with image editing software by virtual extraction of the bone fragments and visually applying optimal fit of local fracture geometry, periosteal and endosteal curvature across adjacent fragments, and internal structures (e.g., canalisation). For the ninth specimen of *Archaeopteryx*, humeral and ulnar parameters were obtained by averaging the values found for two reconstructed circa mid-diaphyseal cross sections each. As the humeral and ulnar geometry of the fifth and seventh specimen are distorted to a markedly larger degree than those of the ninth specimen, they are represented by the single best-preserved cross section present in the circa mid-diaphyseal domain.

The elements of *Compsognathus* and *Rhamphorhynchus* used in this study were recovered from the Solnhofen Plattenkalk as well, and were reconstructed following the same protocol as the *Archaeopteryx* material (Supplementary Fig. 10). One fragment of cortical bone is conspicuously absent at the optimal sample location for the *Compsognathus* ulna in the upper right quadrant of the bone in the extracted slide, as also evidenced by an ulnar cross section extracted 3.58 mm proximal to the used sample location (Supplementary Fig. 10). The geometry of this cortical fragment at the sample location was reconstructed through close

comparison with the bone and fracture geometry visible in the referred more proximal cross section.

All transverse cross sections were converted to binary cortical bone profiles by tracing the periosteal and endosteal surfaces and subsequently filling the area of the original cortical bone white[41]. Occasionally occurring spongy bone and obvious irregularities, such as cracks or protruding splints, were digitally removed. The area of the few small splints in fossil material that could not be accurately repositioned in their exact original orientation was taken into account during restoration of typically the periosteal margin. Cross-sectional geometric parameters were calculated with MomentMacro 1.4 (http://www.hopkinsmedicine.org/fae/mmacro.html) in the public-domain image analysis software ImageJ (https://imagej.nih.gov/ij/).

Most species in our data set are represented by the humerus and ulna of a single adult individual, although some were included as a composite of elements sourced from two individuals, or as average values derived from elements from two, three or four individuals (see Supplementary Data 1). Individuals sampled in this study are believed to represent adults based on element size and bone structure. *Archaeopteryx* specimens are often considered to be juveniles, which has been specifically concluded for the specimens of *Archaeopteryx* included in this study through relative size and bone surface texture[2,13]. The sampled *Compsognathus* specimen was also reported to represent a juvenile individual[42]. The studied *Rhamphorhynchus* individual is of comparably small size, suggesting juvenility as well. Gender composition across the data set is generally unknown and was therefore not considered.

**Locomotor modes and body mass.** Avian flight mode categorisation notoriously suffers from the qualitative, non-discrete nature of faunal flight strategies[20]. To overcome classification-specific effects in discriminant analysis, we considered the classifications suggested by Viscor et al.[21] and Close et al.[20] independently. Both avian flight mode divisions were expanded with one group that encompasses volant wing-propelled diving auks, and supplemented with alternative archosaurian locomotory strategies represented by exemplary taxa (Supplementary Data 1). The avian flight mode categories *sensu* Viscor et al.[21] encompass (1) short flight, (2) forward flapping/bounding flight, (3) high-frequency flapping flight, (4) undulating flight and (5) gliding–soaring flight, which were assigned following the proposed taxonomical designations[21]. Taxa not included in their work were assigned flight modes according to the provided description[21]. *Geococcyx californianus* was classified as 'short flight' rather than as 'forward flapping/bounding flight' proposed for Cuculidae. A second, more recent avian flight mode division by Close et al.[20] separates (1) burst flight, (2) intermittent bounding flight, (3) continuous flapping flight, (4) flap-gliding flight and (5) soaring flight, and was applied through description. We chose to score volant wing-propelled divers separately in both subdivisions as their aquatic locomotory strategy is known to profoundly influence wing bone morphology[17,43] and, consequently, the expression of flight-related adaptations recorded therein[43]. Both referred avian flight classifications were complemented with the following locomotor categories: (6) long-tailed pterosaurian flight, (7) short-tailed pterosaurian flight, (8) (avian) non-volant wing-propelled diving, (9) ratite bipedal, (10) (non-avian) dinosaurian bipedal, (11) (non-avian) dinosaurian omnipedal and (12) crocodilian quadrupedal.

Body mass values for extant taxa were either directly available for the referred individuals or sourced from online databases[44,45] and literature[46,47] as species averages (see Supplementary Data 1). For extinct forms, either specimen-specific body mass estimates[13,39,48] or average specific body mass estimates were available[49,50]. The Malagasy shelduck *Alopochen sirabensis*, reported to have been "slightly larger" than the Egyptian goose *Alopochen aegyptiaca*[51] (average body mass of 1900 g[44]), was assigned a reconstructed body mass of 2000 g. Body mass for the *Rhamphorhynchus* sp. MdC 20269891 was reconstructed through the relation between body mass and wing span for basal pterosaurs proposed by Witton[52]. Total wing length for MdC 20269891 was measured as the cumulative length of the humerus (19 mm), radius (34 mm), wing metacarpal (14 mm), phalange I (47 mm), phalange II (40 mm), phalange III (35 mm) and phalange IV (44 mm) taken from photographic and scan data, and amounts to 233 mm. In the Dark Wing specimen of *Rhamphorhynchus muensteri* (JME SOS 4785; Jura Museum Eichstätt), the distance between left and right glenoid measures 1.56 × humeral length, which proposes an original interglenoid distance of 30 mm for for McD 20269891. Its corresponding wingspan, calculated as twice the wing length plus the interglenoid distance, amounts to 0.496 m. From the relation of Witton[52] follows a reconstructed body mass of 95 g. The body mass for the *Brasileodactylus araripensis* individual in our study was inferred through close morphological and dimensional agreement between its humerus (length 168 mm, maximum distal width of 47) and the humerus of AMNH 22552 (length 170 mm, maximum distal width of 46 mm)[53,54], for which a reconstructed wingspan of 3270 mm was reported[55]. From the described relation between wingspan and body mass in pterodactyloids[56] follows a reconstructed body mass of 6540 g.

Body mass values for the studied specimens of *Alligator mississippiensis* (141 cm[38]) and the domestically bred *Crocodylus niloticus* (200 cm; personal observation PT) were reconstructed through specific allometric scaling relations between body length and body mass offered in literature[57,58].

**Cross-sectional parameters**. Relative cortical thickness[12] (CA/TA) and mass-normalised resistance against torsional forces[12] (J/M) were quantified for archosaurian humeri and ulnae (Supplementary Data 1, Supplementary Figs. 4 and 5). CA/TA describes element hollowness as the ratio of cortical bone area to the total area delimited by the external surface of the bone in cross section (Supplementary Fig. 2). As such, CA/TA is proportionate to the corticodiaphysary index (CDI)[59] and inversely related to the K-parameter[6,10]. Polar moment of inertia of an area J quantifies the mechanical resistance against torsion around the longitudinal axis of the considered element. J mathematically equals the sum of the maximum second moment of area (Imax) and minimum second moment of area (Imin) that quantify resistance against deflection along the respective orthogonal major and minor principal axes (Supplementary Fig. 2) through the relative distribution of matter[12]. Values for J obtained from cross sections with an Imax/Imin >1.50 are typically overestimated[12,60,61], but remain informative when considered proportionally rather than quantitatively (as is its derivative Zp[41,61]). J was normalised over body mass to permit comparison in a highly body mass-diversified comparative framework that spans well over five orders of magnitude (Supplementary Data 1).

Cortical vascular density, expressed as the amount of canals per mm² of bone area in cross-section[62–65], was considered qualitatively for a modest selection of archosaurs for which high-resolution data was available, but not challenged statistically (Supplementary Data 1). Bone area in section was calculated as CA (see 'Cross-sectional geometry' above) with MomentMacro 1.4 in ImageJ 1.49. In the fifth and ninth specimen of *Archaeopteryx*, cortical canals were counted visually. Absolute canal abundance in the cross-sectional cortex of archosaurs other than *Archaeopteryx* was obtained through selection of the darkest grey levels in the greyscale histogram (including canals) by thresholding the cortical domain and subsequently counting the amount of elements within canal size range using the Analyse Particles function of ImageJ 1.49.

The ratio of Imax over Imin provides a reliable measure for the ellipticity of the transverse bone shaft and has been considered as such in biomechanical explorations[12,60,66,67]. These approaches traditionally assume that the degree and orientation of ellipticity reflect an adaptation that offers optimised resistance against bending, with the direction of Imax corresponding to the orientation of the maximum bending moment. However, an opposite functional interpretation of cross-sectional element ellipticity in which a preferred bending direction is achieved through orientation of Imin has also been proposed specifically for avian wing bones[68]. Such conflicting explanations of the same parameter illustrate the complexity of interpreting cross-sectional bone ellipticity in a functional context and thereby obscure the information offered by other characters when assessed in a multivariate context. We therefore chose not to include quantified bone ellipticity measures in our comparative study.

**Tree inference and divergence chronogram**. Mesozoic topology and timing used in this study (Supplementary Data 3 and Supplementary Fig. 1) were derived from the Paleobiology Database (PaleoDB; https://paleobiodb.org) on 29 January 2016. Divergence nodes were adopted as the older bound date for the oldest report of a taxon nested beyond the respective split. Mesozoic terminal nodes and the Tertiary terminal node for *Mancalla cedrosensis* were placed at the younger bound date of their occurrence. *Alopochen sirabensis* is placed at 656 AD, which represents the median calibrated radiocarbon age for the last-occurrence date for the species[69]. The 19th century terminal node for *Pinguinus impennis* was dated through its well-documented last observation in 1844 AD[70]. Topology and timing within the extant avian subset were largely adopted from the well-resolved phylogeny by Jarvis et al.[71] (Supplementary Data 4 and Supplementary Fig. 1), since the more recent neoavian phylogeny proposed by Prum et al.[72] was found to conflict PaleoDB on numerous crucial accounts. Several higher-order divergence times in Aves were obtained from PaleoDB following the procedure described above. Two specific inconsistencies in PaleoDB were negotiated through literature (see Supplementary Note 3). Insufficiently resolvable divergence nodes were placed at a standard + 4 MY with respect to their closest established crownward node. The three *Archaeopteryx* specimens were included as a polytomy at + 4 MY with respect to the older bound date for the genus in recognition of taxonomic and ontogenetic uncertainty[2]. The phylogenetic tree was constructed in Mesquite 3.04[73].

**Statistical analyses**. The relations between individual geometric parameters and locomotor divisions in the training taxa set were statistically assessed through phylogenetic analysis of covariance using the PDAP module of Garland et al.[74]. For each parameter, 10000 unbound simulations were performed along the constructed tree (Supplementary Fig. 1) under a Brownian motion regime in PDSIMUL. ANCOVA was performed with a grouping of the training taxa according to their locomotor classes as response variable, parameter values as predictor variable, and body mass as covariate (Supplementary Data 5).

Phylogenetic PCA[75] scores for the studied taxa, founded on humeral and ulnar CA/TA and J/M (Supplementary Data 2), were obtained with the phyl.pca function (method: BM; mode: cor) of the phytools package[76] in the R-environment[77] through RStudio 0.99.484[78]. The phylogenetic PCA scores were subsequently subjected to Partitioning Around Medoids specified to two clusters with the pam function of the cluster package[79] in RStudio 0.99.484.

The archosaurian groups outside *Archaeopteryx* serve as training taxa that represent known locomotor modes and thus form a morphological reference environment for discriminant analysis. The optimum value of Pagel's λ, the scaling factor of autocorrelation for a certain parameter on a given phylogenetic tree[80] to be applied in phylogenetically informed discriminant analysis, was found using the approach described by Schmitz et al.[81] in RStudio 0.99.484 (Supplementary Fig. 8). Linear discriminant analysis and classification of mystery taxa (Supplementary Data 2) was conducted in PAST 3.10[82], as was one-way MANOVA (Supplementary Tables 1–3) among individual locomotor strategies. Additional motivation for and information on the statistical approach used here is available as Supplementary Note 4.

**Data availability**. All data underlying the study are available in Supplementary Data 1.

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

## Acknowledgements

The authors thank P.E. Ahlberg, V. Fernandez, D.J. Field, D. Hanley, A. Houssaye, G. Oatley, O.W.M. Rauhut, F.J. Serrano, and E.L.R. Simons for discussions or providing advice. D. Berthet (Musee des Confluences, Lyon, France), M. Eriksson (Museum of Evolution, Uppsala, Sweden), C.K. Lefèvre (MNHN, Paris, France) and J.R. Nudds (School of Earth and Environmental Sciences, University of Manchester, UK) are acknowledged for providing access to the material kept in their care. Beamtime was available as inhouse beamtime at the ESRF.

## Author contributions

D.F.A.E.V., P.T. and S.S. conceived the study and designed the experiments, D.F.A.E.V., M.R., V.B., P.T. and S.S. performed the experiments, D.F.A.E.V., J.C., E.d.M. and S.S.

analysed the data, D.F.A.E.V. wrote the manuscript, J.C., E.d.M., M.R., V.B., S.B., P.T. and S.S. helped with and edited the manuscript.

## Additional information

**Competing interests:** The authors declare no competing interests.

