## [Peer Review File · Nature Communications]

Reviewers' comments:

Reviewer #1 (Remarks to the Author):

This is a very interesting paper that tackles some excellent fundamental issues related to the origins of avian flight. I appreciate the use of cross sectional properties from a wide range of living taxa and their comparison to a sample of Archaeopteryx specimens. I think this is a highly valuable approach. The topic is high profile and quite appropriate for Nature Communications.

I do have some concerns with the paper, which are outlined below. I think the methods are mostly sound, but some of the results are difficult to interpret. I do have a few concerns with the methodology for analyzing cross sections, which are noted below, but these are relatively minor. My most pressing concern is with the proposed alternative flight kinematic. While creative and interesting, I don't think proper time and attention has been giving to analysis of the relevant fluid flows and flight dynamics of the proposed wing motions. I almost feel as if this is two manuscripts: one on cross sections, and one on dynamic simulation. It is worth considering splitting them into two papers, which would also give you the space to properly present the kinematic (with associated CFD and/or calculative solutions to some of the flow problems). My specific comments follow below.

Sincerely,

—Michael Habib

Results:

Figure 2 is the primary figure in many ways, and there are a lot of good data presented. However, this figure raised some questions for me regarding the results. On a practical note, it is hard to follow some parts of Figure 2, particularly because some of the color codings are very similar.

On a minor note, "Pterosaurian flight" does not seem like a type of gait or flight performance in the way the other categories are. I realize the contrast here is pterosaurs vs birds, but the term is a bit awkward/unusual. Is there a particular range of flight parameters that pterosaurs are meant to represent? The "pterosaurian flight" category seems to be based mostly, or entirely, on derived pterodactyloid material, especially Anhanguerid material. There is nothing wrong with this, but it also means that the CA/TA values and other variables shown for pterosaurs are among the more extreme examples, and not typical for pterosaurs (see work by L. Martin-Silverstone on this). So perhaps just labeling those points as "pterodactyloids" or "large pterosaurs" would be more sensible.

More importantly, the morphofunctional interpretations (particularly understanding the schematic sequences) are difficult using discriminant functions alone. DF Analysis is useful for distilling several variables into a single morphospace, but the downside is that DA 1 and

DA 2 are not easy to interpret (so, for example, in the "alpha" track leading to Archaeopteryx, it's not altogether clear what the real cross sectional shape changes are along the morphospace path delineated). I think using the discriminant functions here is a fine idea, but a morphospace analysis using the raw variables first would be quite useful for interpretation. Part of what concerns me in this regard is that living "short flight" specialists tend to have humeri with very thin cortices, short total length, and a large diameter (leading to a large value of J). I have cross sectional data from only a single Archaeopteryx specimen (the Munich specimen), so I can't be certain, but the mass-specific J values and humeral:femoral J value ratios for the Munich specimen come out (for me) only roughly in the range of things like burst-flying galliform birds. What you've done here is more sophisticated, of course, by combining J values with CA/TA information in a single morphospace, and that seems to get more signal (which is great). But it that also means there's something important about the relationship between the polar moment of inertia and the CA/TA ratio, and it is hard to interpret what that underlying relationship is in the DA morphospace alone.

Methods:

Overall, the statistical analyses used are appropriate, however, despite the inclusion of a phylogeny, it is not clear how phylogenetic signal was accounted for. Assuming it was, there should still be some discussion of this, especially in CA/TA values (which can be very sensitive to ancestry). There is a brief inclusion of this in noting that alcid have high CA/TA values, but that appears to be the extent of the discussion.

Flight Stroke reconstruction:

This is both quite clever and quite problematic. The suggested flight kinematic (I enjoyed the video sequence very much) seems only to have taken into account range of motion and ignored the actual fluid dynamics involved. I applaud you all for taking the time and effort to try to tackle this problem, but to do so properly requires some serious consideration (and discussion) of how the flows would actually work with the kinematic you have proposed.

Some things to keep in mind (and I apologize in advance if you have already considered these basics): all flying vertebrates produce the majority of their thrust and weight support using a downstroke that essentially "slices" downwards and forwards (i.e. cranially and ventrally). This works because lift is always oriented perpendicular to the direction of flow (which in the case of the wing will be the combination the translational flow as the animal flies forward and the induced flow from the flight stroke). Since the wing moves cranially and ventrally on the power stroke, the resultant lift is oriented mostly upwards (weight support) and somewhat forwards (thrust). The true resultant vector includes drag, of course, which inclines the resultant a bit more caudally, and thus a bit more vertically (more weight support). If the animal is going fast enough, it can use a slight reduction in span on the recovery stroke to continue producing weight support (at the cost of a slight reduction in speed from a bit of negative lift along the craniocaudal axis). For a slow flying animal (so climb out, landing, and short flights - particularly relevant here) the wings need to exit

circulation as completely as possible, shedding the vortices as rings. That means having a recovery stroke that is either highly folded or twists the feathers apart to cut off the bound vortex (modern birds typically use both).

I am having a very hard time understanding how your proposed flight stroke operates, given the above core dynamics. The initial translational stroke would produce lift (given a sufficient angle of attack), and this would provide some weight support. It would, however, also produce negative thrust, as best I can tell. It's also not clear if it would produce sufficient weight support and how the vorticity would be maintained going into the second phase of the flight stroke. The second phase "propulsive" stroke almost looks like a sculling stroke - in other words, it looks like you're proposing a drag-based propulsive stroke, which simply isn't going to work for something as big as Archaeopteryx. Wings are not paddles. Now, I might be missing something here, all we have are the range of motion discussion and the visual of the kinematic, so I'm making my best assessment from those. If you can present a flow analysis that demonstrates the flight stroke would work, then that would be excellent. As it is, I don't think the kinematic is supported, and I would probably split it off to another paper that includes some fluid computations to go with it. At the very least, there needs to be a reasonably supported estimate of the coefficients of lift and drag that would go with each phase of the proposed flight stroke. Basic steady-state estimates with a simplified model would do, but that is a bare minimum.

I also recommend looking at some of the kinematic work (experimental and simulation) by Dr. A. Heers, which suggests that the body position of Archaeopteryx in flight might have been more vertical (this seems to solve many of the shoulder excursion problems). This fits nicely with your short flight hypothesis because short flight behaviors are often at steep angles and use a more vertical body position.

Reviewer #2 (Remarks to the Author):

This manuscript is well written and presents a fairly straightforward hypothesis and analysis. I recommend that it be published with only minor revisions.

My primary concern has to do with their analysis of flight style. Although Archaeopteryx falls among those taxa possessing a "bounding" style, there should be an acknowledgment within the text that, as with any study of this nature, any assumption about flight becomes more tenuous as one moves away from the crown group. Analyses such as this don't position living taxa with 100 percent confidence, and that number only decreases moving down the stem. Stem taxa may have had different forces acting on their wings. For example, the authors describe a completely different flight stroke for Archaeopteryx. What affect would this have on the variables measured in the study?

Also, I don't know if it is logistically feasible, but it would be interesting to include some smaller deinonychosaurs--more in the range of bird size--in the analysis. Right now Deinonychus is the only one included, and it is still a fairly large non-avian dinosaur. If the authors have access to any small troodontids, the addition of these would greatly

strengthen their arguments for flapping flight in Archaeopteryx. The graphs show a clear difference between flying and flightless taxa, but this also correlates pretty well with large and small taxa. If adding more specimens is not feasible, it should be noted that size might have an influence as well.

Reviewer #3 (Remarks to the Author):

Your research into the humeral and ulnar anatomy of Archaeopteryx and your conclusion that this taxa used short-burst, flapping flight is fascinating. Your methods for measuring the specimens, your statistical analyses, and your comparative analyses are carefully and appropriately described, and it appears that your conclusions are robust.

In contrast, your hypothetical flight model (Figure 3, Supplemental Video, Text Lines 138-161) is potentially interesting yet far from adequately formed or rigorous. This is the only major criticism I offer of your manuscript in its present form, but it is serious. Given the lack of adequate development of either muscular or aerodynamic models, it is premature to suggest - as you presently do - that the flapping style of Archaeopteryx was an evolutionary dead end. I suggest you delete this section of your paper, simply condense the discussion to point out the differences in anatomy between Archaeopteryx and later forms that indicate the next challenge is to adequately model muscular and aerodynamic forces to better understand how Archaeopteryx flapped its wings.

Alternatively, it will be necessary to (1) explicitly model muscular forces using something such as Software for Interactive Musculoskeletal Modeling (SIMM) and aerodynamic forces using either a (2a) rough approach of basic hypothesized kinematics modeled using quasi-steady aerodynamic assumptions or (2b) more-sophisticated Computational Fluid Dynamics (CFD).

At present, it is not possible to understand how the flapping stroke you hypothesize can produce weight support and thrust, particularly thrust, as it appears to produce a negative angle of attack of the wing during downstroke. It is unclear what the colors represent in Figure 3 and in the supplemental video. If these are indicating pressure, then perhaps you have performed aerodynamic modeling, but this effort and the relevant conclusions are not described in your manuscript.

Minor comment: Lines 70-72: This sentence is too cryptic, coupled with the numerical citation format, to be adequately clear. What is ontogenetic disparity? Expand slightly on how/why/who suggested Archaeopteryx had a low metabolic rate.

Response to Referees

Our answers (in red) follow the comments of and suggestions by the referees that are included below.

Reviewer #1 (Remarks to the Author):

This is a very interesting paper that tackles some excellent fundamental issues related to the origins of avian flight. I appreciate the use of cross sectional properties from a wide range of living taxa and their comparison to a sample of Archaeopteryx specimens. I think this is a highly valuable approach. The topic is high profile and quite appropriate for Nature Communications. I do have some concerns with the paper, which are outlined below. I think the methods are mostly sound, but some of the results are difficult to interpret. I do have a few concerns with the methodology for analyzing cross sections, which are noted below, but these are relatively minor. My most pressing concern is with the proposed alternative flight kinematic. While creative and interesting, I don't think proper time and attention has been giving to analysis of the relevant fluid flows and flight dynamics of the proposed wing motions. I almost feel as if this is two manuscripts: one on cross sections, and one on dynamic simulation. It is worth considering splitting them into two papers, which would also give you the space to properly present the kinematic (with associated CFD and/or calculative solutions to some of the flow problems). My specific comments follow below.

Answer: We have followed the explicit suggestion provided by both Reviewer #1 and #3 to divide this report into the well-received cross-sectional bone analysis, which we are resubmitting here, and the reconstruction of flight cycle dynamics that will be accommodated in a designated project in the future.

Reviewer #1 (Remarks on Results 1):

Figure 2 is the primary figure in many ways, and there are a lot of good data presented. However, this figure raised some questions for me regarding the results. On a practical note, it is hard to follow some parts of Figure 2, particularly because some of the color codings are very similar.

Answer: We have positioned the legend outside Figure 2 (as well as in the comparable Extended Data Figure 9) to remove distractive clutter. Furthermore, we have assigned abbreviated codes to the group hulls in the plot that correspond with the categories presented in the legend to aid in recognition of group identity on first glance.

Reviewer #1 (Remarks on Results 2):

On a minor note, “Pterosaurian flight” does not seem like a type of gait or flight performance in the way the other categories are. I realize the contrast here is pterosaurs vs birds, but the term is a bit awkward/unusual. Is there a particular range of flight parameters that pterosaurs are meant to represent? The “pterosaurian flight” category seems to be based mostly, or entirely, on derived pterodactyloid material, especially Anhanguerid material. There is nothing wrong with this, but it also means that the CA/TA values and other variables shown for pterosaurs are among the more extreme examples, and not typical for pterosaurs (see work by L. Martin-Silverstone on this). So perhaps just labeling those points as “pterodactyloids” or “large pterosaurs” would be more sensible.

*Answer: We have acquired new comparative data on a more basal, rhamphorhynchoid pterosaur in the size range roughly corresponding to that of *Archaeopteryx*, which has been included in the updated statistical analyses. This did not only add valuable reference to the study as a whole, but also enabled a conservative comparison between the conditions of rhamphorhynchoid and pterodactyloid pterosaurs (included as discrete groups named “long-tailed pterosaur” and “short-tailed pterosaur”, respectively).*

Reviewer #1 (Remarks on Results 3):

*More importantly, the morphofunctional interpretations (particularly understanding the schematic sequences) are difficult using discriminant functions alone. DF Analysis is useful for distilling several variables into a single morphospace, but the downside is that DA 1 and DA 2 are not easy to interpret (so, for example, in the “alpha” track leading to *Archaeopteryx*, it’s not altogether clear what the real cross sectional shape changes are along the morphospace path delineated). I think using the discriminant functions here is a fine idea, but a morphospace analysis using the raw variables first would be quite useful for interpretation.*

Answer: Firstly, we have improved our presentation of the raw data through the inclusion of two bivariate plots that visualise the relations of the considered raw parameters per element (humerus and ulna; see Extended Data Figure 5). Secondly, we have developed our initial discussion of these raw values in the text and included a more elaborate comparison of the raw variables in relation to their locomotory affinity. Thirdly, we have now applied k-means clustering set to two clusters to the raw data set, which returned an identical grouping (principally corresponding to the division between volant and non-volant taxa) as PAM of the phylogenetic PCA scores. Fourthly, in the text we have briefly remarked which raw parameters principally load the discriminant axes recovered. Fifthly, we have abandoned the α - and β -trajectories in the plots. These were aimed at schematically indicating broader trends but should not introduce confusion regarding the particular patterns presented. Such larger-scale pathways remain briefly addressed in the text but do not represent claims that are imperative to the study itself.

Reviewer #1 (Remarks on Results 4):

Part of what concerns me in this regard is that living “short flight” specialists tend to have humeri with very thin cortices, short total length, and a large diameter (leading to a large value of J). I have cross sectional data from only a single Archaeopteryx specimen (the Munich specimen), so I can't be certain, but the mass-specific J values and humeral:femoral J value ratios for the Munich specimen come out (for me) only roughly in the range of things like burst-flying galliform birds. What you've done here is more sophisticated, of course, by combining J values with CA/TA information in a single morphospace, and that seems to get more signal (which is great). But it that also means there's something important about the relationship between the polar moment of inertia an the CA/TA ratio, and it is hard to interpret what that underlying relationship is in the DA morphospace alone.

Answer: We have clarified the valid notion that relative cortical thickness and torsional resistance are partially related more explicitly in the text. However, we care to emphasise that we have focused on relations that are evident *relative* to such a general relation towards identifying the conditions that characterise our groups, to which we have drawn attention in the text as well. Furthermore, we agree that whole-bone geometry, although outside the focus of this study, does indeed importantly contribute to the function of the flight apparatus as a whole. In that light, we realise that *Archaeopteryx* was not capable in engaging in the mode of volancy employed by modern short-flying birds, but conclude that its humeral mid-diaphysal shaft was subjected to a force regime most consistent with that flight strategy in our comparative data set.

Reviewer #1 (Remarks on Methods):

Overall, the statistical analyses used are appropriate, however, despite the inclusion of a phylogeny, it is not clear how phylogenetic signal was accounted for. Assuming it was, there should still be some discussion of this, especially in CA/TA values (which can be very sensitive to ancestry). There is a brief inclusion of this in noting that alcids have high CA/TA values, but that appears to be the extent of the discussion.

Answer: Yes, phylogeny was consistently accounted for in our statistical exploration. Phylogenetic ANCOVA was conducted using PDAP. We furthermore employed phylogenetic Principal Component Analysis through the phytools package in R, and determined the degree of phylogenetic autocorrelation towards Phylogenetically Informed Discriminant Analysis using the R script provided by Schmitz and Motani^{1A}. This latter analysis returned a Pagel's λ of 0.00 for all tested sets used (i.e. each of the locomotory divisions, both with and without mystery taxa), which is now explained in the body of the manuscript itself. However, because we believe their (Schmitz and Motani^{1A}) script for conducting the phylogenetic discriminant analysis itself has some crucial shortcomings, and because phylogenetic autocorrelation is absent towards our tests, Linear Discriminant Analysis was considered the most appropriate way to conduct discriminant analysis^{1B} in this study. LDA it is synonymous with a phylogenetic discriminant approach that uses a Pagel's λ of 0.00. We now incidentally remark on particularly evident cases of phylogenetically shared conditions (e.g. elevated CA/TA in Charadriiformes and elevated J/M in larger continuously flapping wild anatids) in the text. However, we feel that our grouping itself is broadly sampled towards providing a representative reference environment for

contextualising *Archaeopteryx*, and such phylogenetic circumstances are too specific to address individually and go beyond the scope of the study. Moreover, since phylogenetic autocorrelation is demonstrably negligible for our purposes, we believe such descriptions would not contribute to the robustness of the reference morphospace used to assess the condition of *Archaeopteryx*.

Reference 1A: Schmitz, L., & Motani, R. Nocturnality in dinosaurs inferred from scleral ring and orbit morphology. *Science* **332**, 705-708 (2011).

Reference 1B: Hall, M. I., Kirk, E. C., Kamilar, J. M., & Carrano, M. T. Comment on “Nocturnality in Dinosaurs Inferred from Scleral Ring and Orbit Morphology”. *Science* **334**, 1641 (2011).

Reviewer #1 (Remarks on Flight Stroke Reconstruction):

This is both quite clever and quite problematic. The suggested flight kinematic (I enjoyed the video sequence very much) seems only to have taken into account range of motion and ignored the actual fluid dynamics involved. I applaud you all for taking the time and effort to try to tackle this problem, but to do so properly requires some serious consideration (and discussion) of how the flows would actually work with the kinematic you have proposed.

Some things to keep in mind (and I apologize in advance if you have already considered these basics): all flying vertebrates produce the majority of their thrust and weight support using a downstroke that essentially “slices” downwards and forwards (i.e. cranially and ventrally). This works because lift is always oriented perpendicular to the direction of flow (which in the case of the wing will be the combination the translational flow as the animal flies forward and the induced flow from the flight stroke). Since the wing moves cranially and ventrally on the power stroke, the resultant lift is oriented mostly upwards (weight support) and somewhat forwards (thrust). The true resultant vector includes drag, of course, which inclines the resultant a bit more caudally, and thus a bit more vertically (more weight support). If the animal is going fast enough, it can use a slight reduction in span on the recovery stroke to continue producing weight support (at the cost of a slight reduction in speed from a bit of negative lift along the craniocaudal axis). For a slow flying animal (so climb out, landing, and short flights - particularly relevant here) the wings need to exit circulation as completely as possible, shedding the vortices as rings. That means having a recovery stroke that is either highly folded or twists the feathers apart to cut off the bound vortex (modern birds typically use both).

*I am having a very hard time understanding how your proposed flight stroke operates, given the above core dynamics. The initial translational stroke would produce lift (given a sufficient angle of attack), and this would provide some weight support. It would, however, also produce negative thrust, as best I can tell. It’s also not clear if it would produce sufficient weight support and how the vorticity would be maintained going into the second phase of the flight stroke. The second phase “propulsive” stroke almost looks like a sculling stroke - in other words, it looks like you’re proposing a drag-based propulsive stroke, which simply isn’t going to work for something as big as *Archaeopteryx*. Wings are not paddles. Now, I might be missing something here, all we have are the range of motion discussion and the visual of the kinematic, so I’m making my*

best assessment from those. If you can present a flow analysis that demonstrates the flight stroke would work, then that would be excellent. As it is, I don't think the kinematic is supported, and I would probably split it off to another paper that includes some fluid computations to go with it. At the very least, there needs to be a reasonably supported estimate of the coefficients of lift and drag that would go with each phase of the proposed flight stroke. Basic steady-state estimates with a simplified model would do, but that is a bare minimum.

I also recommend looking at some of the kinematic work (experimental and simulation) by Dr. A. Heers, which suggests that the body position of Archaeopteryx in flight might have been more vertical (this seems to solve many of the shoulder excursion problems). This fits nicely with your short flight hypothesis because short flight behaviors are often at steep angles and use a more vertical body position.

Answer: We thank you for elaborately detailing your thoughts on this. Since Reviewer #3 also explicitly advised to split the referred part off to a future study, we have decided to follow up on this suggestion and have removed the detailed presentation of the proposed flight kinematic from this report.

Reviewer #2 (Remarks to the Author 1):

My primary concern has to do with their analysis of flight style. Although Archaeopteryx falls among those taxa possessing a "bounding" style, there should be an acknowledgment within the text that, as with any study of this nature, any assumption about flight becomes more tenuous as one moves away from the crown group. Analyses such as this don't position living taxa with 100 percent confidence, and that number only decreases moving down the stem. Stem taxa may have had different forces acting on their wings. For example, the authors describe a completely different flight stroke for Archaeopteryx. What affect would this have on the variables measured in the study?

Answer: We indeed aimed to interpret the results from the perspective of forces acting on the wing bones conservatively, and therefore concluded our Results section with the notion that the presented results mainly illustrate an affinity with incidental flapping flight in general rather than with a particular modern avian flight mode in particular. To emphasise this, we have expanded the sentence presenting the "relatively low percentages of correctly classified training species" with "this uncertainty undoubtedly carries over to the characterisation of extinct taxa with unknown locomotory strategies". Importantly, this does not influence the well-supported conclusion that *Archaeopteryx* was volant, and that it shares important conditions with birds that employ flapping-based flight, which we also present as the main outcome in the abstract.

Reviewer #2 (Remarks to the Author 2):

Also, I don't know if it is logistically feasible, but it would be interesting to include some smaller deinonychosaurs--more in the range of bird size--in the analysis. Right now Deinonychus is the

only one included, and it is still a fairly large non-avian dinosaur. If the authors have access to any small troodontids, the addition of these would greatly strengthen their arguments for flapping flight in *Archaeopteryx*. The graphs show a clear difference between flying and flightless taxa, but this also correlates pretty well with large and small taxa. If adding more specimens is not feasible, it should be noted that size might have an influence as well.

Answer: Although troodontids were not readily available, we have followed your suggestion and expanded our data set with the small coelurosaur *Compsognathus* for this purpose. We furthermore included *Rhamphorhynchus* to assess the condition of a roughly *Archaeopteryx*-sized basal pterosaur. Finally, we visualised and studied the anterior limb bone geometry of the new, bird-like maniraptoran species *Halszkaraptor escuilliei* that we describe in Nature this week (49). Unfortunately, its diaphyseal shafts experienced a substantial yet unquantifiable degree of secondary dissolution, as also stated in that report, which prevents reliable quantification of its original mid-diaphyseal bone architecture. In the revised manuscript, we have also clarified that a size effect is present, and that we focus on signals *relative* to this size effect when assessing the locomotor modes under consideration.

Reviewer #1 (Remarks to the Author 1):

Your research into the humeral and ulnar anatomy of Archeopteryx and your conclusion that this taxa used short-burst, flapping flight is fascinating. Your methods for measuring the specimens, your statistical analyses, and your comparative analyses are carefully and appropriately described, and it appears that your conclusions are robust.

In contrast, your hypothetical flight model (Figure 3, Supplemental Video, Text Lines 138-161) is potentially interesting yet far from adequately formed or rigorous. This is the only major criticism I offer of your manuscript in its present form, but it is serious. Given the lack of adequate development of either muscular or aerodynamic models, it is premature to suggest - as you presently do - that the flapping style of Archeopteryx was an evolutionary dead end. I suggest you delete this section of your paper, simply condense the discussion to point out the differences in anatomy between Archeopteryx and later forms that indicate the next challenge is to adequately model muscular and aerodynamic forces to better understand how Archeopteryx flapped its wings.

Alternatively, it will be necessary to (1) explicitly model muscular forces using something such as Software for Interactive Musculoskeletal Modeling (SIMM) and aerodynamic forces using either a (2a) rough approach of basic hypothesized kinematics modeled using quasi-steady aerodynamic assumptions or (2b) more-sophisticated Computational Fluid Dynamics (CFD).

At present, it is not possible to understand how the flapping stroke you hypothesize can produce weight support and thrust, particularly thrust, as it appears to produce a negative angle of attack of the wing during downstroke. It is unclear what the colors represent in Figure 3 and in the supplemental video. If these are indicating pressure, then perhaps you have performed

aerodynamic modeling, but this effort and the relevant conclusions are not described in your manuscript.

Answer: We have followed the explicit suggestion provided by both Reviewer #1 and #3 to divide this report into the well-received cross-sectional bone analysis, which we are resubmitting here, and the reconstruction of flight cycle dynamics that will be accommodated in a designated project in the future.

Reviewer #1 (Remarks to the Author 2):

Minor comment: Lines 70-72: This sentence is too cryptic, coupled with the numerical citation format, to be adequately clear. What is ontogenetic disparity? Expand slightly on how/why/who suggested Archeopteryx had a low metabolic rate.

Answer: We have elaborated on this aspect in the manuscript following your suggestions while avoiding ambiguous terminology and clarifying the context and claim presented.

REVIEWERS' COMMENTS:

Reviewer #1 (Remarks to the Author):

This is one of the most thorough responses/rebuttals to a review I have ever received, and it is greatly appreciated. I particularly note the substantial amount of work that went into improving the results section, and this section is greatly improved. You have effectively addressed all of my concerns, and my comments below are mostly thoughts for future work (for what that's worth). I also have included a suitably stoic academic apology for being picky about your phylogenetic corrections, when you had actually accounted for this issue quite effectively:

Your comments and rebuttal regarding my concerns on phylogenetic autocorrelation are well taken. I clearly underappreciated your corrections in this regard (i.e. I read too quickly), and I appreciate that description of the Schmitz and Motani script has been added to the main body of the paper. You are, of course, correct that since there is demonstrably negligible phylogenetic autocorrelation in your dataset, the more extended description I suggested in the prior review is moot.

Splitting the paper into two parts (leaving the gait simulation for a future publication) was a good move; I look forward to seeing more on the reconstructed flight gait in the future.

I was particularly pleased to see the addition of a rhamphorhynchoid pterosaur in the revised analysis, which I think greatly strengthens the bird to pterosaur comparisons. I was somewhat surprised at the relative polar modulus results for the Rhamphorhynchus specimen, and I am quite intrigued.

I highly encourage you to follow up on this with a broader sample of rhamphorhynchoid specimens in the future, as their mechanics are sorely understudied compared to that of pterodactyloid taxa. Applying the same techniques to a comparative rhamphorhynchoid sample (especially with more age classes represented) would make for a very interesting paper.

Reviewer #2 (Remarks to the Author):

The authors have addressed all of my concerns with the original manuscript. Although they were not able to add a small deinonychosaur, they did add a compsognathid and addressed issues having to do with size. I recommend that this study be published.